# Features of Psychomotor Coordination in Adolescents with Neuropsychiatric Pathology Enrolled in a Standard Educational Program [note 1]

**DOI:** 10.3390/brainsci12020245

**Published:** 2022-02-10

**Authors:** Polina Mavrenkova, Natalia Pankova, Marina Lebedeva, Mikhail Karganov

**Affiliations:** Institute of General Pathology and Pathophysiology, 8 Baltiyskaya St., 125315 Moscow, Russia; dubynina.polina@yandex.ru (P.M.); nbpankova@gmail.com (N.P.); ma_lebedeva@mail.ru (M.L.)

**Keywords:** psychomotor tests, mental disorders, adolescents, educational process, monitoring

## Abstract

The imbalance between the speed and accuracy of cognitive-motor operations can lead to the formation of abnormal behavioral programs fraught with serious negative consequences for the individual. For successful correction and prevention of social disadaptation in adolescents with nervous and mental diseases and functional disorders in mental sphere in general education schools, the peculiarities of their psychomotor activity should be taken into account. We measured some parameters of visual-motor coordination and sensorimotor reaction in adolescents with mental disorders with (n = 36) or without (n = 27) organic brain damage. Adolescents from both groups showed higher speed, but poorer accuracy and smoothness, of movements than typically developing students (n = 70). The visual and acoustic reaction times were longer in adolescents with mental disorders without organic brain damage than in reference groups.

## 1. Introduction

According to statistics, more than 2.2 billion of the world’s population are children and adolescents, among which 10–20% of young people with neuropsychiatric disorders (in some countries up to 39%) [1,2]. In a half of adult patients with severe forms of mental disorders, mental problems appeared before the age of 15 [3]. Epidemiological studies show a threefold increase in prevalence rates of mental disorders in children and adolescents between 1990 and 2016 years. A recently developed epigenetic, transgenerational model predicts a further increase in the number of mental illnesses due to epigenetic influences (nutrition, microbiota, overweight, etc.). This makes it possible to carry out preventive measures in order to reduce the likelihood of psychopathology [4]. The World Health Organization has noted the need to create living conditions and environments for individuals with mental problems to increase their chances of improving mental health [5]. In this context, the inclusive education system, along with other supportive measures for people with mental health problems [6], can be considered a promising model for reducing their stigmatization and integrating them into society [7]. However, it should be taken into account that these children can have serious problems in general education school due to some peculiarities of neuropsychic processes (attention, social communication, motor control, and executive functions) [8,9]. In adolescence, adaptation to school life can be even more complicated due to peculiarities of this age period [10]. Schools can be a serious source of stress and anxiety, which can reduce academic achievements.

Psychomotor disturbances were demonstrated for many mental disorders, even in cases when they are not among the main diagnostic (clinical) symptoms of the disease [11,12,13,14,15]. Changes at all levels of organization of motor acts (sensory input, neural networks, muscle reaction) can affect the behavior of an individual. It is believed that motor dysfunctions can appear at the early stages of ontogeny in individuals who later develop a psychopathology, in particular, psychotic reactions, and can represent a premorbid characteristic of these states [16]. However, the development of motor skills is a dynamic process. In the childhood, motor skills rapidly progress, and motor deficits in children with neuropsychiatric problems are quite noticeable. In adolescence, the development of motor skills is decelerated [17], which can mask deviations in the rate of maturation of the neuromotor circuits in adolescents with mental disorders, and they do not differ from children with typical development by many parameters of fine motor skills and coordination [18].

Schoolchildren with mental problems obviously need medical supervision, as well as psychological and pedagogical support. For assessing physiological and mental functions of students under conditions of educational institutions, the use of relatively simple methods for monitoring of their functional state is advisable [19].

Our objective was to study the features of psychomotor coordination in adolescents with neurodevelopmental disorders by using simple motor tests.

## 2. Materials and Methods

### 2.1. Participants

The study was carried out as part of a municipal program to develop a technology for express assessment of the health status of the population based on a comprehensive polysystemic examination [20], including, among other indicators, an assessment of the state of psychomotor systems. The data were obtained in the process of sanogenetic monitoring of the health status of schoolchildren in Moscow in educational institutions included in this project.

We included in the analysis data obtained during testing of adolescents aged 11.9–17.9 years—students of a specialized school for patients of psychiatric clinics at the stage of remission, as well as students referred by a psychoneurologist as in need of psychological and pedagogical rehabilitation (“Psychiatric patients”, n = 63). Based on medical records, the participants from this school were divided into two groups. The first group consisted of persons whose diagnoses were associated with nosological category “Organic mental disorders”—F00–F09 according to ICD-10 (“Organic disorders patients”, n = 36). They were predominantly adolescents diagnosed with F06 (mental disorders due to brain damage and dysfunction and to physical disease, n = 27) and F07 (personality and behavioral disorders due to brain disease, damage and dysfunction, n = 9) (Table 1). The second group included subjects with mental disorders without a history of organic brain pathology (“Without Organic disorders patients”, n = 27): F20–29 (schizophrenia, schizotypal and delusional disorders, n = 6), F30–39 (mood disorders, n = 4); F40–49 (anxiety disorders, n = 4); F80–89 (mental developmental disorders, n = 7); and F90–99 (emotional and behavioral disorders beginning in childhood and adolescence, n = 6) (see Table 1 for detail). The control group (Control, n = 70) included typically developing adolescents—the students of general education schools, randomly selected from a large array (more than 5000 persons) of surveyed schoolchildren without a psychiatric diagnosis in such a way that the distribution of the sample by age corresponded to the distribution in “Psychiatric patients” group (Figure 1).

Most commonly, adolescence is divided into three developmental periods: early adolescence (10–14 years of age), late adolescence (15–19 years of age), and young adulthood (20–24 years of age) [21], according to recommendations of WHO and a Lancet commission on adolescent health [22]. Many variables associated with sensorimotor function are not yet fully developed in early adolescence, and the mechanisms that provide sensorimotor coordination continue to mature throughout adolescence [23]. Considering this fact, we divided the surveyed cohort of schoolchildren into two age ranges in accordance with the mathematical rounding rules: early (11.9–14.6 years) and late (14.7–17.9 years) adolescents. The sex and age characteristics of the examined samples are presented in Table 1.

Informed consent for participation in the study was obtained from all students and their parents (or legal representatives) in accordance with the Universal Declaration on Bioethics and Human Rights (articles 5, 6, and 7). Compliance with international and Russian legislative acts on the legal and ethical principles of conducting scientific work involving human subjects was confirmed by the decision of the Ethics Committee of the Research Institute of General Pathology and Pathophysiology (Protocol No. 1, 22 January 2019).

### 2.2. Equipment

For assessing the psychomotor coordination in adolescents, a computerized device for psychomotor diagnostics (INTOX LLC, St. Petersburg, Russia) was used (Figure 2A).

### 2.3. Examination Procedure

During one session, the subjects performed two tests, each involving right-hand and left-hand movements (all subjects were right handed). During the first test “Basic motor reactions”, the examinee moved the lever in a cyclic mode (right–left) within the range marked by external LED markers. The examinee was instructed to move the lever from one LED marker to the other with maximum possible speed and accuracy (i.e., to change the direction of movement strictly in the LED-marked point). After the examinee acquired a certain rhythm of movement, the presented light markers were suddenly changed (the external markers were switched to the internal pair and, after a while, again to the external pair). In accordance with the preliminary instructions, the subject had to change promptly the amplitude and direction of movement (change of the motor stereotype).

During the second test, “Simple Sensorimotor Responses”, the subject was instructed to move the lever from the designated mark (right or left external LED for the left and right hands, respectively) in response to an acoustic or light stimulus and then quickly and accurately return the lever to its original position. The duration of the acoustic and light stimuli was 0.28 and 0.4 s, respectively; in each series, 10 stimuli were presented with randomly generated interstimulus intervals (2–5 s).

The following parameters were evaluated: duration of the movement cycle (in ms) was measured as the mean time of lever movement from one marker to another and back; time to change the motor stereotype (in ms) was measured as the time to achieve the required accuracy of movement in the new amplitude mode; the error of sensory correction of flexors (in %) and extensors (in %) was determined as the ratio of the mean deviation from the specified movement range boundaries to the total amplitude of lever movement for the entire cycle; smoothness of movement (in%) is the contribution of the main harmonic in the Fourier spectrum of the movement, motor asymmetry (in%) (Figure 2B), and the time of a simple sensorimotor reaction to visual and acoustic stimuli (in ms) measured as the time from stimulus to the beginning of lever movement from the initial point (Figure 2C). A detailed description of the device and the formula for calculating the estimated indicators is given in the article [24].

### 2.4. Statistical Analysis

We used Statistica 7 (StatSoft Inc., Tulsa, OK, USA) and GraphPad Prism 6 (GraphPad Software Inc., San Diego, CA, USA) software for the analysis and graphical presentation of data.

We checked all variables for a normal distribution using the Kolmogorov–Smirnov test. According to the results of the preliminary verification, we rejected the hypothesis of normal data distribution for the most of the studied parameters. Hence, we used a nonparametric unpaired Mann–Whitney U test for independent variables (two-sided) for comparative analysis with false discovery rate (FDR) control [25] to correct for multiple comparisons, Wilcoxon signed-rank test and Spearman Rank Order Correlations. The effect size was calculated using an online program on the website Psychometrica [26]. Kohen’s coefficients are given in Appendix A. The results are presented as the median and interquartile range (Me [Q1; Q3]).

## 3. Results

### 3.1. Indicators of Sensorimotor Coordination and Simple Sensorimotor Reaction in Adolescents with Typical Development and in Adolescents with Mental Disorders

Comparison of the test results for the right and left hands in each group revealed differences both in “Control” and “Psychiatric patients” only for the time to change the motor stereotype (Appendix A). To simplify the analysis, the values of other parameters were averaged for both hands.

When comparing the values of the estimated indicators in males and females, no significant differences were found in any parameter in “Control”. In “Psychiatric patients”, females demonstrated longer Movement cycle duration (*p* = 0.034, M-U test) and more Smoothness of movement (*p* < 0.001, M-U test) than males.

The correlation analysis revealed associations of some parameters of sensorimotor reactions with age in “Control”: the Movement cycle duration, the Time of the reaction to acoustic stimulus, and the Motor asymmetry decreased; the Error of correction of flexors and extensors increased (Table 2). In “Psychiatric patients”, the correlation with age was statistically significant only for the Motor asymmetry, and there was no correlation with age for other indicators.

Comparison of the psychomotor coordination parameters with considering the selected age ranges (see Table 1), showed that the values of all these indicators (with the exception of Motor asymmetry), both in early and late adolescents of “Psychiatric patients” group, differ from the corresponding indicators of “Control” (Figure 3). “Psychiatric patients” demonstrated higher speed (decrease of the Movement cycle duration and Smoothness of movements in comparison with “Control”) but lower accuracy of motor reactions (increase of the Error of correction of flexors and extensors). There were no significant differences in these indicators between early and late adolescent in “Psychiatric patients” with the exception of a single indicator—only Motor asymmetry in late adolescents was less than in early ones. In late adolescents of “Control”, the Movement cycle duration decreased, while Error of correction of extensors increased in comparison with the early adolescents (see Figure 3).

Parameters of simple sensorimotor reactions in “Psychiatric patients” were higher than in “Control”: Time of the reaction to visual stimulus—both for early and for late adolescents; Time of the reaction to acoustic stimulus—only for late adolescents. In “Control”, the Time of the reaction to visual stimulus was less for late adolescents than for early ones. In addition, although the response time to an acoustic stimulus was not statistically different in the age subgroups of “Control”, the ratio (Time of the reaction to acoustic stimulus/Time of the reaction to visual stimulus) was smaller in older adolescents (U = 344, Z = 2.221, *p* = 0.025). This ratio did not differ between early and late adolescents in “Psychiatric patients” (U = 348, Z = 0.665, *p* = 0.514), nor between the corresponding age subgroups in “Control” and “Psychiatric patients” (U = 150, Z = 1.047, *p* = 0.305 and U = 1023, Z = −0.446, *p* = 0.660, respectively).

### 3.2. Indicators of Sensorimotor Coordination and Simple Sensorimotor Reaction in Adolescents with Typical Development and in Adolescents with or without Organic Brain Pathology

Analysis of the parameters of psychomotor coordination in “Control”, “Organic disorders patients”, and “Without organic disorders patients” revealed only differences between healthy adolescents and both groups of students with mental disorders (Figure 4). No differences were found between “Organic disorders patients” and “Without organic disorders patients”. On the contrary, the Time of reactions to visual and acoustic stimuli in “Without organic disorders patients” was longer than in “Control” and “Organic disorders patients”. No differences were found between the “Control” and “Organic disorders patients” in the Time of reaction to visual and acoustic stimuli. All groups did not differ in ratio (Time of the reaction to acoustic stimulus/Time of the reaction to visual stimulus).

## 4. Discussion

The study revealed differences in indicators of psychomotor coordination and sensorimotor reactivity in performance brief yet comprehensive motor tests in typically developing adolescents and adolescents with mental disorders studying according to a standard educational program.

We examined sex differences and age effects of youths when performing simple motor tasks. There are evidences that males tend to outperform females in tests of speed and accuracy of movement [27,28,29]. However, we found no sex differences the parameters of psychomotor activity in adolescents with typical development. Perhaps the lack of sex differences in the speed and accuracy of performing motor tests in our study is due to their simplicity. Other researchers have shown, for a battery of simple motor tasks, that two sexes differed only marginally in speed but markedly in associated movements [30]. The speed of simple movements can equalize due to asynchronous somatic and endocrinological maturation in males and females. According to another hypothesis, faster and more accurate movements in males are associated not with the initial motor characteristics but with the better learning of males in motor skills. This advantage is enhanced during adolescence [31]. In the subsequent analysis, we considered pooled groups of males and females.

In typically developing adolescents, a correlation was found between the speed of movements with age (the duration of cyclic movements decreased), which is consistent with the results of other authors. In a study by our colleagues using the same motor tasks, it was shown that, in healthy children aged 6 to 16 years, the development of motor functions occurs non-linearly, which, apparently, is associated with a change in the physiological mechanisms of regulation of movements. Up to 9–10 years, the leading role in the implementation of movements belongs to visual control. The morphofunctional development of the frontal areas of the cortex determines the transition to movement control based on pre-formed motor programs. The maturation of the mechanism of central commands is practically completed by the age of 14. Late adolescents exhibit the same type of motor control as adults [32]. The hypothesis that the improvement in speed and accuracy indicators with age is associated with the function of the prefrontal lobe is confirmed in a cohort study (more than 3 thousand participants) of the neurocognitive development of people aged 8–21. The tests included sensorimotor and motor tasks. The speed itself also improved with age, even in the absence of a cognitive task. The increase in speed was more pronounced for a simple motor task than for a task requiring sensorimotor integration. Although the sensorimotor cortex and cerebellum are among the first to mature [33], performance continues to improve in early adulthood. It is possible that ongoing maturation of the brain influents on motor function [28]. On the other hand, there is evidence that adolescence is a period of delay or regression in at least some of the sensorimotor mechanisms in children (see review in Reference [23]).

In typical development late adolescents, an increase in the speed of movements is accompanied by a decrease in accuracy (an increase in the error in the correction of extensors). For adolescents with mental disorders, we found no sex- or age-related effects on psychomotor coordination and sensorimotor reactivity. However, differences from typical develop adolescents were observed for almost all parameters in both age subgroups: the speed of movement was higher, and error correction in accurate placing the cursor on the light mark was worse than in the corresponding control subgroups. The data obtained in the study that the late adolescents showed less accuracy in performing cyclic movements, at first sight, does not agree with the literature data. A number of studies have shown that the accuracy of movements in children and adolescents increases with age [28,30], However, in most cases, tests are used that measure only this indicator: the subjects solve only one task (for example, hit the target with a ball in physical testing or point the cursor at the desired point on the screen in neurocognitive research). In our study, the subject was instructed to move the lever as quickly and as accurately as possible.

Movement cycle duration can be interpreted as an indicator of functional mobility of nervous processes that provide a change in the direction of movement of the hand [34]. Cyclic movements consist of two phases: ballistic, mediated by excitation in the nervous system, and correcting, that provides accurate adjustment of the movement to the target mediated by the inhibitory processes. The Smoothness of movement and Error of correction of extensors and flexors reflect the balance of the processes of excitation and inhibition in the nervous system the lower is Smoothness of movement, the greater is the imbalance between the excitation and inhibition; excessive extension/flexion may be associated with inefficient functioning of the inhibitory systems in the motor areas of the brain. The correction is based on information received through the visual and proprioception feedback. The time and intensity of inhibition of the motor reaction are determined, first of all, by anticipation of the future result of the action. The neural mechanisms underlying movement planning and control are now extensively studied in humans [35] and in animal model experiments [36].

In the other side, according to test 1 paradigm, cyclic movements are performed under visual control, i.e., we evaluated visual-motor coordination during performance of a simple motor task. Visual signals coming with the dorsal stream to the parietal cortex [37] are necessary for planning and implementing the targeted action, as well as for processing the information about movement. Programming of the movement and feedback through the kinesthetic and visual channels are necessary for successful everyday activities. Different components of fine motor activity in typical development schoolchildren are formed with different rates, which is clearly seen from the profiles of maturation of visual-motor coordination during the ontogeny that were determined by some widely used tests [38]. The authors showed that some parameters of visual-motor reactions do not further improve since late childhood or early adolescence, while some psychomotor functions continue to mature in late adolescence, and even in early adulthood.

However, in our opinion, based on the observation of the behavior of adolescents during testing, late adolescents, given the choice between speed and accuracy, prefer speed to the detriment of accuracy, while younger teenagers try to follow instructions. Recently, there has been a lot of interest in the idea that the variability of sensorimotor behavior depends not only on noise, redundancy, adaptability, learning, or plasticity of sensory and motor components but also on unique individual strategies, particularly from the style of sensorimotor behavior typical of human group, context, or period [39].

In “Psychiatric patients”, Smoothness of movement is reduced in comparison with the “Control”. Impaired coordination (including visual-motor coordination) is one of the most common motor dysfunctions in individuals with neuro-psychiatric problems. Deviant perceptual-motor coordination is believed to be involved in the development of social deficits in children and adolescents with autism spectrum disorders [40]; it can be one of the most robust characteristics of the endophenotype of individuals at risk of developing schizophrenia [41], and it is associated with impulsive behavior in bipolar disorder and borderline personality disorder [42]. Patients with severe depressive disorder demonstrate a deficit of executive control, which, under multitasking conditions, manifested in impairment of cognitive and sensorimotor functions [43].

In our study, “Organic disorders patients” and “Without organic disorders patients” did not differ by the parameters of visual-motor coordination of movements. Taking into account heterogeneity of the “Psychiatric patients”, it can be hypothesized that the observed disorders of psychomotor coordination are unspecific and can be associated with a wide range of neuropsychiatric disorders. Visual-motor dysfunction can be related to impaired functioning of neural networks of attention, executive control, sensory and motor zones of the cortex, and/or impairment of their interaction. The heterogeneity of the clinical sample, as well as its small size, is certainly a limitation of this study. Recently, using stepwise linear regression on adolescent males with mental illness, it was shown that a significant part of the variance in motor ability scores falls on diagnostic categories [44]. Overall, the clinical group performed significantly worse than the control group on all motor tests.

It has been demonstrated that the time of sensorimotor reactions depends on the experimental conditions [45]. In particular, movement limits, i.e., the need to stop movement at a certain point, increases the latency of the response. In our study, movements in response to visual and acoustic stimuli did not require high accuracy and were not constrained by the amplitude; we tested only the speed of motor reaction that, in fact, reflects the speed of pulse conduction in central nervous system. This process involves perception and processing of the sensory signal and central organization of the motor act culminating in bioelectric activation of the muscles. It has been previously demonstrated that preparation for and initiation of the movement are mechanically independent, and each act has a distinct neural basis [46]. Detailed analysis of the components explains why the reaction time is usually slower than possible: movement initiation is delayed relative to the mean time needed for preparation to avoid the risk of movement initiation before it is properly prepared. Even in highly stimulus-oriented tasks, stimulus presentation does not directly cause movement. More likely, the stimulus triggers an internal decision whether to make a move, thus reflecting voluntary rather than reactive mode of control. In the “Psychiatric patients”, Time of the reaction to visual stimulus was higher than in the “Control”: in late adolescents—significant, in early—tend to increase. These findings are consistent with the data of other authors who reported an increase in the time of simple sensorimotor response in adolescents with mental disorders, unhealthy psychological symptoms, and emotional problems [47,48,49]. We also showed that “Without organic disorders patients” showed greater Time of the reaction to visual and acoustic stimulus and in comparison with “Organic disorders patients” and “Control”. In a study on patients of different ages (7–79 years old) in a psychiatric clinic using the Tapping test, all examined spontaneous ultradian cycles were shown, in frequency and amplitude of which healthy and only non-organically ill patients differed from patients with an organic neurological deficit [50]. It is likely that residual organic brain disorders in adolescents in remission do not significantly affect the time of simple motor reaction. There are published data that the time of simple sensorimotor reactions in subjects with organic brain injuries (for example, after traumatic brain injury) during recovery period did not differ from the control values [51].

## 5. Conclusions

One objective of this study was to explore the features of motor reactions in adolescents with mental disorders in remission when performing simple motor tasks. In accordance with our expectations, participants from the clinical population showed significant differences from typically developing peers in almost all assessed indicators of psychomotor coordination and sensory-motor reactivity. Sensory and psychomotor deficits have been identified in psychiatric patients in both early and late adolescence. Another objective of this study was to evaluate to what extent a diagnosis—in particular, the presence or absence of a history of organic brain disorders—contributes to variation in motor functions. Significant differences between these subgroups were revealed only in the latency of a simple sensorimotor reaction to light and sound stimuli. The disorders of psychomotor coordination are unspecific and can be associated with a wide range of neuropsychiatric disorders. Results of this study indicate that the motor abilities of adolescents with mental disorders is in need of attention, regardless of diagnosis. We suggest using simple motor tests as an affordable and cheap method for monitoring of the functional state of schoolchildren with neuropsychiatric problems.

## Figures and Tables

**Figure 1 brainsci-12-00245-f001:**
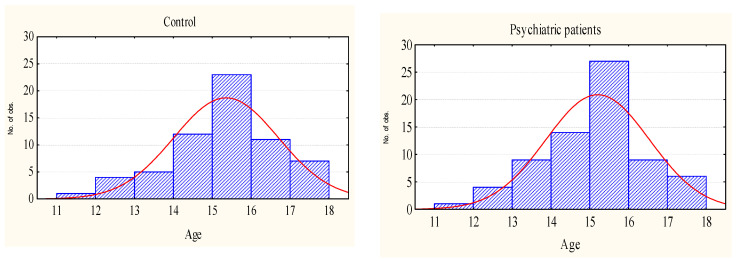
Age distribution of adolescents of the examined groups.

**Figure 2 brainsci-12-00245-f002:**
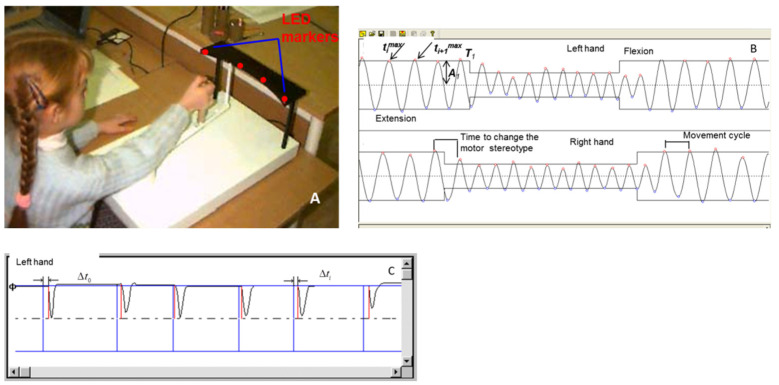
(**A**): The device records the trajectory of hand movement during simple motor tests. The forearm of the subject lies on the platform, and the hand holds a vertical stick of a lever that can move in a horizontal plane (movement “from the elbow”). The vertical stick of the lever with a cursor at the end moves between two pairs of LED markers located on an arc-shaped panel. The angular distance between the outer pair of LED markers relative to the axis was 70° and between the inner pair 35°. (**B**): An example of a kinematogram for the right and left hands during the “Basic motor reactions” test. (**C**): An example of a record obtained during the “Simple Sensorimotor Responses” test.

**Figure 3 brainsci-12-00245-f003:**
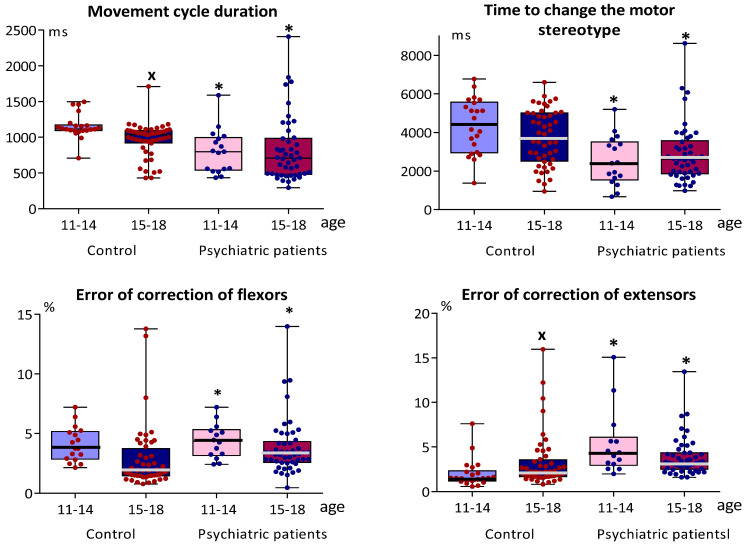
Parameters of psychomotor coordination and simple sensorimotor reaction in the early and late adolescents of “Control” (n = 22 and n = 48, respectively) and “Psychiatric patients” (n = 17 and n = 46, respectively) The median is indicated by a bold line, column boundaries: Q1, Q3 (first and third quartiles), spread: minimum and maximum values; * *p* < 0.01 < *p*_cr_ in comparison with the corresponding age subgroup of “Control” ^x^
*p* < 0.05 in comparison with the early adolescent of the corresponding group.

**Figure 4 brainsci-12-00245-f004:**
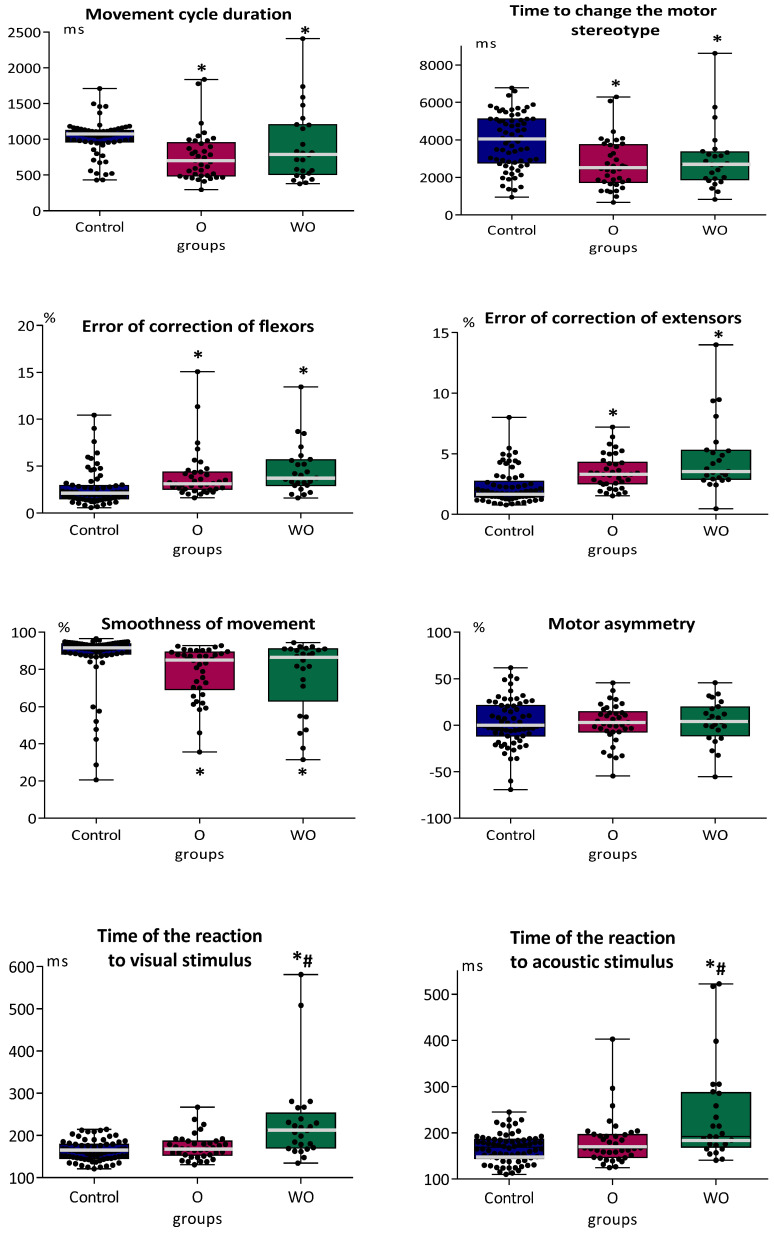
Parameters of psychomotor coordination and simple sensorimotor reaction in adolescents with mental disorders with a history of organic pathologies of the brain (O, n = 36), without organic pathology (WO, n = 27), and typically developing adolescents (“Control”, n = 70). The median is indicated by a bold line, column boundaries: Q1, Q3 (first and third quartiles), spread: minimum and maximum values; * *p* < 0.05 in comparison with “Control”; # *p* < 0.01_;_ in comparison with “Organic disorders patients” (O).

**Table 1 brainsci-12-00245-t001:** Sex and age characteristics of adolescents in the examined groups.

Group	Control (Typical Development)				Psychiatric Patients							
					Without Organic Pathology of the Brain				With Organic Pathology of the Brain			
Sex	Female		Male		Female		Male		Female		Male	
Age	Early	Late	Ealy	Late	Early	Late	Early	Late	Early	Late	Early	Late
n	8	24	14	24	3	8	4	12	3	4	7	22
ICD-10 codes/n					F43.2/1 F84.0/1 F91.9/1	F20.0/1 F21/1 F32.0/2 F40.8/1 F42.8/1 F98.5/2	F84.0/3 F20.0/1	F21/1 F29/2 F30.1/1 F31.0/1 F43.2/1 F84.1/1 F84.9/1 F89/1 F91.9/1 F92.8/1 F98.5/1	(F06.7/1; F07.8/1 F07.9/1)	F06.6/1 F06.8/2 F07.8/1	F06.6/3 F06.7/1 F07.0/2 F07.8/1	F06.6/14 F06.7/2 F06.8/3 F07.0/3

Note to Table 1: F06.0 Organic hallucinosis; F06.3 Organic mood (affective) disorders; F06.6 Organic emotionally labile (asthenic) disorder; F06.7 Mild cognitive disorder; F06.8 Other specified mental disorders due to brain damage and dysfunction and to physical disease; F07.0 Organic personality disorder; F07.8 Other organic personality and behavioral disorders due to brain disease, damage and dysfunction; F07.9 Unspecified organic personality and behavioral disorder due to brain disease, damage and dysfunction; F20.0 Paranoid schizophrenia; F21 Schizotypal disorder; F29 Unspecified nonorganic psychosis; F30.1 Mania without psychotic symptoms; F30.9 Manic episode, unspecified; F31.0 Bipolar affective disorder, current episode hypomanic; F31.1 Bipolar affective disorder, current episode manic without psychotic symptoms; F32.0 Mild depressive episode; F40.8 Other phobic anxiety disorders; F42.8 Other obsessive-compulsive disorders; F43.2 Adjustment disorders; F84.0 Childhood autism; F84.1 Atypical autism; F84.9 Pervasive developmental disorder, unspecified; F89 Unspecified disorder of psychological development; F91.9 Conduct disorder, unspecified; F92.8 Other mixed disorders of conduct and emotions; F98.5 Stuttering (stammering). ICD-10 codes/n—the number of patients with the specified diagnosis.

**Table 2 brainsci-12-00245-t002:** Nonparametric Spearman coefficients of correlation (r) of psychomotor coordination parameters with the age of the examined adolescents.

Parameters of Movement	Control	Psychiatric Patients
Movement cycle duration	−0.54 *	−0.11
Time to change the motor stereotype	−0.19	0.01
Time of the reaction to visual stimulus	0.05	0.07
Time of the reaction to acoustic stimulus	−0.27 *	−0.04
Error of correction of flexors	0.27 *	0.02
Error of correction of extensors	0.43 *	0.07
Smoothness of movement	−0.18	−0.08
Motor asymmetry	−0.26 *	−0.31 *

Note: * *p* < 0.05, Spearman Rank Order Correlations.

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
