# Peer review of "Features of Psychomotor Coordination in Adolescents with Neuropsychiatric Pathology Enrolled in a Standard Educational Programâ€"

_brainsci, 2022, doi:10.3390/brainsci12020245_

Round 1
Reviewer 1 Report
The authors compared several parameters of psychomotor coordination in adolescents with diverse psychiatric disorders to typically developing peers. The authors also compared individuals with psychiatric disorders to individuals with organic brain disorders. Correlations between psychomotor function and age were also examined. The authors report differences between patients and controls across several psychomotor functions, as well as differences between patients with and without organic brain damage. Associations between psychomotor function and age were also reported, mostly in controls. The analyses are thorough, but several aspects of the paper lack clarity.
“while young people with neuropsychiatric disorders in some countries constitute up to 39% (on average 10-20%)” – 39% of what? All young people?
“In the childhood, motor skills rapidly progress and motor deficits in children 43 with neuropsychic problems are quite noticeable” – should neuropsychic be neuropsychiatric?
There are many abbreviations throughout the text, which make it a little hard to follow. I am not sure that “early adolescence” and “late adolescence” need to be abbreviated. Similarly, I would not abbreviate the group names - patients and controls would be clearer. What do groups O and P refer to on page 5, lines 159-164?
How were patients and controls recruited into the study?
The Ns for all groups are presented by sex and age in table 1, but I think the total N for each group (controls, psychiatric patients, organic disorder patients) should be presented in the text too, definitely in the abstract.
I appreciate that codes are listed in table 1 but the disorders should also be named, for ease. How many individuals have each of these diagnoses? There are 63 patients, but only 19 diagnoses listed so seemingly some appear in the sample multiple times. Given the heterogeneity of the patient group, it would be helpful to know whether the majority have a developmental disorder, or psychosis, or an affective disorder.
Why was 14.6 used as the cutoff for the age groups? It also means that the late adolescence group is quite a lot bigger than the early adolescence group.
I found the description of the motor task quite hard to follow. Higher resolution and more photos would help illustrate the different tests and parameters measured. I would also appreciate a table with all the parameters, their abbreviations and what they measure. I do not understand the EFC and EEC parameters. SM and MA are also not adequately explained.
The statistical analysis section does not outline all the analyses that were conducted.
I find it a little unusual to examine sex differences with correlations. The same is also true of age, regression could be used to examine both age effects and sex differences and would give more information (beta, se, p).
The authors only report statistics for the TCMS because this is the only parameter that showed a significant difference between patients and controls, but statistics (and standardized effect sizes) should be presented for all parameters in a table, regardless of statistical significance.
The authors examine age effects in two ways – with age as a continuous variable and then by comparing early and late adolescence – why? The authors also seem to assume that age effects are linear – is this the case?
As stated in the beginning of the results section the TCMS is the only parameter to show a significant difference between patients and control, but figure 3 shows significant group differences for both early and late adolescence for MCD, EFC, EEC, and TRV – why is this? It seems there may be an age by group interaction effect – did the authors test for this?
I suggest using one significance level for the figures, the multiple symbols for group vs. age effects and then also for different significance levels make them quite busy.
Are different parameters presented in figure 4? There are only 6 panels compared to the 9 in figure 3. Also, should TSMS be TCMS? The psychomotor parameters should be presented in the same order everywhere – text, tables, figures. It would make it easier to interpret all the results.
The effects presented in figure are interesting – the parameters that show differences between patients and controls do not show differences between the two patient groups, and vice versa. Does this suggest that some psychomotor functions are related more to general pathology while others are related more to brain damage?
Why are the results and discussion one section? Subheadings in the results section would also be helpful.
Reviewer 2 Report
I have some considerations to improve your work. The study sample is not very heterogeneous. It is important to specify in the participants section which mental disorders are involved, which are the organic problems referred to. Furthermore, it would be advisable to understand if they are subjects who have already received a diagnosis and made psychodiagnostic evaluations. It is appropriate to add in the text what are the strengths, the limits and add future perspectives. Reorganize the article by providing a section dedicated exclusively to results. Please, amplify the bibliography.
Author Response
Please see theattachment

Round 2
Reviewer 1 Report
Thank you for your clarifications – the methods and results are much clearer.
Some of the phrasing remains a little confusing throughout e.g.:
“There were no significant differences in these indicators between early and late adolescent in “Psychiatric patients”. Only Motor asymmetry in late adolescents was less than in early ones.”
The first sentence seems to say there are no differences and then the second sentence says there are differences.
I now understand that the first section of the results is simply reporting on differences between the right and left hands and table 2 could probably be in the supplement rather than in the main text.
However, a table with results for all statistical tests done is still missing, a few statistics are presented in the text, but most are not. It is therefore unclear how significant these findings are. Effect sizes (standardized mean differences i.e., Cohen d) would also be useful since the sample is quite small.
The reasoning behind the early and late adolescence remains a little unclear to me – the authors say in the introduction that differences are bigger in childhood than adolescence, do they also predict differences between early and late adolescence? I still think it would be interesting to test for group by age interactions since there is an effect of age in controls, but not patients, potentially suggesting slowed or halted development in patients. I read the Pankova et al. paper the authors reference regarding the cutoff of 14.6, but the cut off in this paper seemed to be in accordance with WHO i.e., 15. Perhaps this is a minor detail, but it seems illogical to lower the cutoff when most of the sample are above the age of 15 and the early adolescence group is much smaller than the late adolescence group.
Reviewer 2 Report
Thanks to the authors for making substantial changes in the text. The disorders of psychomotor coordination are unspecific and can be associated with a wide range of neuropsychiatric disorders. An important limitation of this study (as highlighted) is the heterogeneity and small size of the clinical sample. Please add a section "conclusion" of the study.
